# Development and Testing of a Daily Activity Recognition System for Post-Stroke Rehabilitation

**DOI:** 10.3390/s23187872

**Published:** 2023-09-14

**Authors:** Rachel Proffitt, Mengxuan Ma, Marjorie Skubic

**Affiliations:** 1Department of Occupational Therapy, University of Missouri, Columbia, MO 65211, USA; 2MathWorks, Inc., Natick, MA 01760, USA; mengxuan.mary.ma@gmail.com; 3Department of Electrical Engineering and Computer Science, University of Missouri, Columbia, MO 65211, USA; skubicm@missouri.edu

**Keywords:** stroke, rehabilitation, algorithm, ambient monitoring, depth sensors

## Abstract

Those who survive the initial incidence of a stroke experience impacts on daily function. As a part of the rehabilitation process, it is essential for clinicians to monitor patients’ health status and recovery progress accurately and consistently; however, little is known about how patients function in their own homes. Therefore, the goal of this study was to develop, train, and test an algorithm within an ambient, in-home depth sensor system that can classify and quantify home activities of individuals post-stroke. We developed the Daily Activity Recognition and Assessment System (DARAS). A daily action logger was implemented with a Foresite Healthcare depth sensor. Daily activity data were collected from seventeen post-stroke participants’ homes over three months. Given the extensive amount of data, only a portion of the participants’ data was used for this specific analysis. An ensemble network for activity recognition and temporal localization was developed to detect and segment the clinically relevant actions from the recorded data. The ensemble network, which learns rich spatial-temporal features from both depth and skeletal joint data, fuses the prediction outputs from a customized 3D convolutional–de-convolutional network, customized region convolutional 3D network, and a proposed region hierarchical co-occurrence network. The per-frame precision and per-action precision were 0.819 and 0.838, respectively, on the test set. The outcomes from the DARAS can help clinicians to provide more personalized rehabilitation plans that benefit patients.

## 1. Introduction

Stroke is the leading cause of serious long-term disability in the United States, and nearly 800,000 individuals in the United States experience a stroke each year [1]. The physical impairments occurring from stroke include hemiparesis, spasticity, fatigue, and pain, and an estimated 65–70% of individuals with stroke experience upper extremity dysfunction [1]. These impairments and dysfunctions significantly impact areas of daily life such as bathing, dressing, feeding, grooming, sports and leisure, and paid or unpaid work [1]. Rehabilitation providers can address limitations in function and participation through evidence-based rehabilitation interventions. However, high demands are placed on the clinician’s time, and the resources (e.g., insurance payments) are limited.

To address many of these gaps in insurance coverage and demands on therapists’ time, most therapists prescribe home programs for their clients with stroke [2]. The practice of activities and exercises at home is crucial for rehabilitation. The adherence rates to self-guided exercise programs in the home setting are notoriously low [3,4,5] and very difficult to quantify due to the reliance on patients’ subjective feedback and reliance on patients maintaining accurate records of their exercise sessions in an exercise diary. Researchers have explored technology-based approaches to increasing adherence to home programs [6]. These approaches include virtual reality and augmented reality games [7], robotics [8], and mobile apps [9]. Positive outcomes have been demonstrated [6,7,8,9], although few studies report long-term gains or sustained levels of activity. Additionally, rehabilitation clinicians are only able to track performance or measure outcomes when the individual post-stroke is using the device or sensor.

### Previous and Related Work

In the past decade, researchers have explored a variety of methods for assessing activity and measuring performance contextually. Approaches with the use of RFID [10,11,12], smartphones with inertial and geospatial capabilities [13,14,15], and inertial sensors [16,17,18,19,20,21,22,23] have been explored.

However, inertial sensors present some limitations. First, although they work fine for global body motion activity types, they fail to provide competitive performance when used alone for local interaction activities such as eating, food preparation, and house cleaning. This is because they only capture two domains of features (time domain and frequency domain), which is not sufficient for local interaction activity types [24]. To enrich the features extracted from these activities, researchers fuse the information from inertial sensors with that from other sensors. Given that camera-based sensors can provide contextual information, many researchers fuse the inertial data with camera-based data [25]. Nam et al. [26] and Hafeez et al. [27] fused accelerometer features with camera-based features to increase the classification accuracy for ambulatory activities. Doherty et al. [28] used the context provided by cameras to identify the specific classed of activity once an accelerometer had identified the level of activity being undertaken. Meng et al. [29] collected acceleration, angular velocity, and surface electromyography data synchronously from 5 upper-limb-worn sensor modules to evaluate the upper-limb Brunnstrom Recovery Stage (BRS) via three typical ADLs (tooth brushing, face washing, and drinking). Second, several studies have revealed that using a single accelerometer might not be sufficient for activity classification [24,30]. Researchers address the problem by combining the data from multiple accelerometer and gyroscope sensors [31,32,33,34]. Third, another disadvantage of using inertial sensor is that they need to be worn all the time. As a result, the sensors must be lightweight and small enough to provide the highest degree of comfort and convenience [35]. Fourth, inertial sensors may drift away from their ideal waring positions during movements, which causes a loss of contact with the skin. Thus, the accuracy and quality of the data may not be stable. Last, inertial sensors are usually difficult to use in cases of long-term monitoring due to their low battery capacities [35,36].

Activity recognition using video-based methods has grown in popularity with advances in the field of computer vision. With the development of computing ability and the improvement of sensor techniques, various data modalities including RGB data, depth data, and skeletal data have been introduced. Depth sensors have the ability to sense the 3D visual world and capture low-level visual information; thus, the depth data for a person can be extracted more easily and accurately [37]. Skeletal joints encode the 3D joint positions of a person. Since the movements of the human skeleton can distinguish many actions, it is promising to exploit skeletal data for action recognition [37].

To recognize an action from a given video, features are extracted and encoded to represent the input video. The encoded features are processed by a classifier to output the class of the action [38]. During the last two decades, a large number of novel approaches for activity recognition have been proposed for both feature extraction and classification. For feature extraction, the three-dimensional corners, blobs, and junctions are representations that can be extracted from a video segment using the spatiotemporal interest points (STIPs) method [39,40]. A large set of gradient-based descriptors have appeared for action recognition, such as histogram of oriented gradients (HOG) [41,42], cuboid descriptor [43], and scale-invariant feature transform (SIFT) [44] techniques. Dense trajectory (DT) [38,45] features were introduced as a form of descriptors that track the path of motion; the improved dense trajectory (iDT) [46] feature has achieved great performance among hand-crafted features. In recent years, there has been a surge of algorithms relying on convolutional neural networks (CNN), which can also be used for feature extraction [38]. For classification, support vector machines (SVMs) and hidden Markov hodels (HMM) have been widely used to provide classification decisions. A fuzzy rule-based system was also designed to learn activities of daily living [47].

Before CNN-based algorithms became popular for action recognition and detection, iFV-encoded iDT features with HOG, HOF, and MBH descriptors using a linear SVM classifier were the top-performing hand-crafted features, achieving accuracy rates of 57.2% and 85.9% on the HMDB51 and UCF101 datasets, respectively [38,48,49]. The classification accuracy was increased to 94.6% by using deep-learned convolutional features.

Since most realistic action-related videos are untrimmed with sparse segments of interest, action recognition itself is insufficient for analyzing actions in real-life videos, as it requires as inputs trimmed action segments. Recently, researchers have started investigating temporal action localization on the THUMOS’14, MPII Cooking Activities and MPII Cooking 2 Activities, and Activity Net datasets [38], as action recognition and localization from untrimmed videos have been demanded in many scenarios such as activity monitoring in an unstructured home environment.

The objectives of temporal activity localization are to localize the temporal boundaries of actions and to classify the action categories simultaneously. Previous studies have investigated the temporal action localization task in full and limited supervision settings based on the level of action annotations, including supervised learning and weakly supervised learning [50].

In supervised learning temporal action localization, the temporal boundaries and action category labels of action instances are needed for each untrimmed video of the training set. The inference goal is to predict the temporal boundaries and the action labels of each action instance [50]. The prediction is based on temporal proposal generation methods, which can be categorized as anchor-based and anchor-free approaches [50]. The anchor-based approaches generate dense multi-scaled temporal proposals and extract proposal features with the same length for each proposal, such as 3D RoI pooling in the R-C3D algorithm [51]. TSA-Net [52] employs a multi-tower network and achieves better performance compared with 3D RoI pooling methods.

Weakly supervised temporal action localization usually requires only the video-level labels of actions during training. During testing, both temporal boundaries and action categories are predicted. The most common approach to weakly-supervised temporal localization is to use an attention mechanism to focus on discriminative snippets and combine salient snippet-level features into a video-level feature [50]. The attention scores are utilized to localize the action boundaries and eliminate background frames. Attention signals are predicted with class-specific attention and class-agnostic attention methods [50]. Temporal localization examples with class-specific attention include UNet [53], action graphs [54], and BaSNet [55]. Temporal localization examples with class-agnostic attention include STPN [56] and BG-modeling [57]. The approaches for the depth videos described above were trained and tested on simulated datasets with pre-designed actions; most actors in the studies were positioned in the environment in a specific location at a specific angle to the depth sensor. Additionally, no studies have investigated action recognition in the stroke population.

Our team has used depth sensors in elderly populations in TigerPlace, an aging-in-place facility, and assisted living facilities with good success for fall detection and gait monitoring, starting in 2011 [58,59]. Building on this work, we developed a Kinect-based system for daily activity recognition and assessment (DARAS). The first version of the system used a Kinect V2 sensor connected to a Windows-based system for data collection and processing. We collected training data from 10 healthy subjects in a mock kitchen. The dataset included 28 actions in 5 categories. We applied the HON4D [42] algorithm to recognize actions from manually segmented depth videos in the first version [60]. In the second version, an ensemble network was used to detect and localize actions in untrimmed depth videos containing continuous unsegmented actions. The ensemble network consists of a customized convolutional–de-convolutional Network (CDC) [61], a customized region convolutional 3D network (R-C3D) [51], and the proposed region hierarchical co-occurrence network (RHCN). Using this method, the CDC and R-C3D can localize actions in continuous depth video streams, and the RHCD can localize actions in continuous skeletal joint frames. The prediction results from these three sub-networks were fused to fully utilize the collected data as different types. Furthermore, this approach outperforms state-of-the-art methods in per-frame action labeling. With success in these two small studies in healthy individuals, we moved to using the Foresite Healthcare depth sensor with individuals post-stroke. Therefore, the purpose of this study was to develop, train, and test a home-based, ambient clinical tool for individuals post-stroke called the Daily Activity Recognition and Assessment System (DARAS). In this paper, we focus on the daily activity recognition methods and results. The assessment component is described elsewhere [62].

## 2. Materials and Methods

There are two main parts of the daily activity recognition system: part one is the action data logging system and part two is the action recognition and temporal action localization component. We explored different depth sensors to determine the most efficient and convenient way to collect daily motion data. Two versions of action recognition algorithms were explored and evaluated to provide accurate action recognition results from manually segmented videos to real-life unsegmented video streams.

### 2.1. Action Data Logging System

The action logging system in DARAS records depth and skeletal data. Depth data are made of pixels that contain the distance from the camera plane to the objects. Skeletal data are a series of entries of 3D Cartesian points, specifying the locations of joints in 3D space during the recorded time.

The Foresite Healthcare system is a standalone system [63]. The hardware modules, including an Astra depth camera, processor, Wi-Fi module, and memory, have been integrated into one small unit. An action logger application was developed to initialize and utilize the Astra SDK to retrieve the depth data and joint data. The depth data are matrices with depth values, which represent the distances of the objects in the view to the depth sensor. The joint data are a list of joint locations of the detected persons in the view in three dimensions. After retrieving the data, the logger application converts the raw depth data to a format that can be easily interpreted. Finally, the logger saves the depth data and joint data to a memory space.

The Astra SDK provides a low-level depth stream and higher-level joint body stream. The depth stream contains depth data from the sensor. The data array included in each DepthFrame contain values in millimeters for each pixel within the sensor’s field of view. The body data are computed from the depth data. They contain the 2D and 3D positions of 19 joints and floor information for up to five different individuals.

The sample rate of the depth data is about 30 frames per second. In the initial version of the action logger application, the raw depth data were converted to grey-level pixel values and each frame was converted to an image frame in .png using ImageMagick library V7. Finally, the .png frames were saved in the memory space. Unfortunately, the saving rate for the depth data was about 3 frames per second. After tracking the processing time for each function in the initial version, the two most time-consuming processes were converting a pixel matrix to a .png image and saving the images to the memory due to the large image sizes.

To increase the frame rate above 6 frames per second, we optimized the process. First, we conducted .png image conversion during post-processing. Then, we compressed the depth matrices to reduce the writing time of the depth data to the internal memory. Lastly, we implemented a multi-thread saving mechanism to write the depth matrices to memory. Specifically, a customized data structure named frame_data was created to store a depth matrix and a customized data structure named write_cache was created to save a batch of frame data wrapped with a header and tail for validation. The size of write_cache was set to 30 frames. The status variable was also included in write_cache to indicate the writing status of this cache. The main process looks for an empty status in write_cache and writes the depth matrix into the cache. While writing the cache, the status of that cache is changed to ‘consume’. Whenever the cache is full, the status of the cache will also be updated to ‘full’. Multiple threads have been created to handle the process of writing cache data to memory. Whenever a cache’s status changes to full, a thread will compress the data in the cache and write the compressed data to memory. The semaphore mechanism was applied to organize the threads. With this optimization, the frame rates of the depth frame and joint data were increased to at least 8 frames per second.

Several mechanisms were included to ensure self-recovery, such as a restart with email notification to the research team, monitoring by the operating system to ensure that the logger process was running, and monitoring of memory usage with email notifications. The data logging system and a sample of collected data are shown in Figure 1.

### 2.2. Action Recognition Algorithm

In our prior research, we explored the most appropriate sensor setup location and angle that would ensure the optimal collection of data for action recognition [60,64,65]. Additionally, we developed an initial set of actions and tested early versions of the algorithm on healthy individuals. For the first version of the system, we collected training data from 10 healthy subjects in a mock kitchen. The dataset included 28 actions in 5 categories, including washing, meal preparation, gadget manipulation, general picking tasks, and walking. We applied the HON4D [42] algorithm to recognize actions from manually segmented depth videos in the first version [60]. In the second version, an ensemble network was proposed to recognize and localize actions in untrimmed depth videos and skeletal joint frames containing continuous unsegmented actions. In the ensemble network, the prediction was fused by the learning from three different sub-networks, which were the convolutional–de-convolutional network (CDC) [61], region convolutional 3D network (R-C3D) [51], and region hierarchical co-occurrence network (RHCN). Using this method, the ensemble network can localize actions in continuous depth and skeletal joint frames. Further, it outperforms state-of-the-art methods in per-frame action labeling. These actions were walking, sitting, reaching above the head, reaching forward, reaching below the waist, hand manipulation, sweeping, and “none of the above”. In this second version, we tested the algorithm on three healthy individuals performing a simple cooking task in their own home kitchens. The average per-frame and per-action accuracy rates were 83.1% and 83.3%, respectively.

#### 2.2.1. Study Participants

Participants were eligible to participate in the study if they (1) were 18 years or older, (2) had experienced a stroke of any sequalae, (3) had difficulty using at least one of their arms during everyday tasks, and (4) were able to ambulate with or without an assistive device. Participants meeting the first two criteria were identified via the Stroke Registry at the University of Missouri. Screening of the second two criteria was performed via a phone call. All participants reviewed and signed an informed consent form before enrolling in the study. This study was reviewed and approved by the Institutional Review Board at the University of Missouri (IRB #2017864).

#### 2.2.2. Data Collection and Processing

The Foresite depth sensor was installed in the home of each study participant for a period of 3 months. The sensor was installed in the kitchen in an unobtrusive location. The location varied for each study participant based on the layout of their kitchen and the places at which they performed the majority of kitchen tasks. For example, one participant had a long galley-style kitchen, with most appliances and the sink all on one side. The depth sensor was installed under the upper cabinet at about the chest level of the participant. The depth sensor then provided a side view of most tasks completed at the stove, sink, and refrigerator. Basic demographic data were collected during the installation visit, including the participant’s age, gender identity, handedness pre-stroke, and side affected by the stroke.

The Foresite system compresses the binary depth data (see Figure 2). For algorithm training and testing, we uncompressed the binary depth data and converted the files to .png frames. The .png frames were then rotated and filtered to ensure no more than one person was in view.

##### Data Annotations

The per-frame action labels are required to train the proposed ensemble network, and the subject identification labels for each action segment are necessary for conducting kinematic assessments for individuals within their homes.

To develop the ground truth for the algorithm, each frame of a subset of data was labeled by a trained research assistant. The action categories for labeling are listed below:Walking;Reaching overhead;Reaching forward;Reaching below the waist;Hand manipulation;None of the above.

Short action segments (with lengths less than 5 frames) were removed from the dataset. Long “none of the above” segments (with lengths greater than 150 frames) were shortened to 50 frames.

In this study, we focused on single-person action recognition. Consequently, frames involving multiple individuals were filtered out during the labeling process. We defined a sequence of motions as a ‘reaching action,’ which begins with the raising of an arm and concludes with the lowering of the arm. The segments where multiple action categories performed simultaneously were labelled as none of the above, such as walking while reaching. A final review of the labels was conducted to ensure consistency in defining the start and end frames across the entire dataset.

For the kinematic assessment of stroke individuals, an individual identification label is required. Once the per-frame action labels were prepared, the action segments, along with their corresponding action category labels, were generated. The identification label was manually assigned to each action segment.

After we labeled each frame, the number of frames for each action category was counted. Data augmentation was performed on less common action categories, and the “none of the above category” was down-sampled to balance the dataset.

#### 2.2.3. Action Recognition Algorithm

##### HON4D Descriptor

Having a robust worldwide representation of a series of depth images holds significance to differentiate similar actions from each other. Several kitchen activities exhibit contrasting actions, primarily distinguished by the reversal of movements in a temporal context, such as the opening and closing of different objects. By incorporating temporal data, it becomes feasible to create a histogram of the regular 4D algorithm (HON4D) [42]. We developed an innovative variant that involves utilizing the initial half of the HON4D descriptor creation process, excluding the addition of projectors to the histogram that has already been computed [42].

The initial stage of the methodology involves computing normals for each pixel within a provided collection of depth images I = {i_1_, i_2_, ···, i_k_}, as in Algorithm 1. The elements comprising the normals represent the variations in depth, as summarized below:(1)$$n=\left(\frac{\partial z}{\partial x},\frac{\partial z}{\partial y},\frac{\partial z}{\partial t},-1\right).$$

After calculating all normals they are normalized, as only their directions are relevant for the bin contributions. Following this, a polychoron is uniformly initialized in the 4D space, where the vertices are regarded as vectors and referred to as projectors, serving as bins for the histogram [42]. The contributions are calculated as the dot product of every normal and projector, as in Algorithm 2. Following the computation of each dot product and its subsequent addition to the appropriate histogram bin, a normalization process is applied.
**Algorithm 1.** **Pseudocode used to generate a list of oriented 4D normals for a sequence of images.**1:   procedure CALCULATENORMALS(images)2:    for k = 0; k < images.Count – 1; k++ do3:      img1 ← images[k]4:      img2 ← images[k + 1]5:      for x = 0; x < img1.Width; x++ do6:        for y = 0; y < img1.Height; y++ do7:          currentPixel = img1.GetPixel(x, y)8:          nextPixel = img2.GetPixel(x, y)9:          rightPixel = img1.GetPixel(x + 1, y)10:          leftPixel = img1.GetPixel(x − 1, y)11:          upPixel = img1.GetPixel(x, y − 1)12:          downPixel = img1.GetPixel(x, y + 1)13:    x = rightPixel – leftPixel14:    y = downPixel – upPixel15:    z = currentPixel – nextPixel16:           normalList.Add(x, y, z − 1)17:        end for18:      end for19:    end for20:    return normalList21: end procedure

**Algorithm 2.** Pseudocode used to generate a histogram of oriented 4D normals, where proj is the list of projectors, normalList is the list of normals calculated from Algorithm 1, and hon4d is the histogram.1: procedure CREATEHON4D(proj, normList, hon4d)2:  for k = 0; k < proj.Count; k++ do3:    for n = 0; n < normList.Count; n++ do4:      hon4d[k] += max(0, dotP(proj[k],norm – List[n]))5:    end for6:  end for7:  return *hon*4*d*8: end procedure

To improve the distinctiveness of a HON4D descriptor, it is essential to divide the image sequence into smaller cells [42]. The Kinect sensor captures depth data at a 512 × 424 resolution. Our cells have dimensions of 4 × 4 × 3 (width × height × depth). As normals are calculated, they are assigned to their respective cells. When all normals are allocated, an individual HON4D descriptor is computed for each cell. Subsequently, these descriptors are combined and subjected to normalization, resulting in the creation of a histogram with 120 bins.

#### 2.2.4. Ensemble Network Architecture

It is a challenging task to recognize and localize the clinically relevant actions from a realistic environment. Three networks that outperformed the others in the RGB dataset were selected and customized to the depth-based collected datasets. To ensure a more accurate prediction outcome, an ensemble network was proposed to fully utilize the collected data from different data types. The network includes three networks, which are the 3D convolutional–de-convolutional network, region convolutional 3D network, and region hierarchical co-occurrence network.

##### Convolutional–De-Convolutional (CDC) Network

Shou et al. [61] proposed a convolutional–de-convolutional (CDC) network that places CDC filters on top of 3D ConvNets. The CDC network performs spatial down-sampling to extract the action semantics and temporal up-sampling to preserve the time information for each frame. Thus, it provides the prediction score for each frame, which can be used to locate the actions.

Convolution neural networks (CNN), in which the convolution kernel is two-dimensional, have been widely used in image classification, detection, segmentation, and other tasks. For video analyses, the temporal features need to be preserved. However, 2D convolution processes cannot capture the timing information very well. Therefore, 3D convolution neural networks (3D CNN), which consists of 3D ConvNets followed by three fully connected layers, were proposed in [66]. Although the 3D CNN can learn the advanced semantic abstraction information of time and space, the output of the video time sequence length is decreased by 8 times. Thus, the fine-grained time information is lost.

For timing location problems, the timing output should be consistent with the input video, although the output size should be reduced to 1 × 1. Motivated by pixel-level semantic segmentation, Shou et al. [61] proposed a CDC filter that generates two 1 × 1 points for each input feature map. As a result, the filter performs the convolution in space (for semantic abstraction) and de-convolution in time (for frame level resolution) simultaneously. It is unique in jointly modeling the spatial–temporal interactions between summarizing high-level semantics in space and inferring fine-grained action dynamics in time.

The input video segment size is 112 × 112 × L, a continuous L frame 112 × 112 image. After the C3D network, L is sampled down to L/8 in the time domain, and the image size in space is sampled from 112 × 112 to 4 × 4. Then, the time domain is sampled up to L/4 in CDC6 and the image size is continuously down-sampled to 1 × 1 in the spatial domain. The time domain is sampled up to L/2 in CDC7. Next in CDC8, the time domain is sampled up to L, and 4096 × K + 1 is used in the full connection layer, where K is the number of classes. The last layer is the SoftMax layer. The final output is (K + 1, L, 1, 1), where K + 1 stands for K action categories plus the background class.

The CDC network was evaluated using THUMOS’ 14, an untrimmed RGB sport action video dataset. The evaluation results show that the model outperforms state-of-the-art methods in video per-frame action labeling. Due to the privacy requirement, a network that can perform temporal action localization on depth kitchen action videos is desired in DARAS. However, the proposed CDC network was designed for RGB videos. Therefore, we first adopted the CDC network for depth videos and then fine-tuned the network using a new collected depth video dataset.

A piece of untrimmed depth video, as shown in Figure 3, is input into the CDC network, in which the 3D convolution neural network is used to extract semantics, and the CDC network is used to predict the dense frame number level scores. Since a depth image only has one grey channel compared to an RGB image, the input of the network is adjusted for depth videos. The time boundary of action instances is identified by grouping the same labels of frames.

##### Region Convolutional 3D Network

The region convolutional 3D network (R-C3D) [51] recognizes and detects actions from untrimmed continuous videos. The key innovations include effectively extracting spatiotemporal features using the 3D ConvNet [66] and extending the 2D RoI pooling in Faster R-CNN to 3d RoI pooling to extract features from proposals with various lengths. The R-C3D network consists of three components: a shared 3d ConvNet feature extractor, a temporal proposal subnet, and an activity classification and refinement subnet. Both spatial and temporal features are essential for representing action sequences. To ensure the action recognition and localization processes are accurate, it is important to extract meaningful spatiotemporal features. A 3D ConvNet encodes rich spatiotemporal features by extending the 2d convolutional layer to 3d, and the temporal information is also preserved while learning the spatial information. Specifically, the features are learned from the convolutional layers (conv1a to conv5b) of C3D. The conv5d activations are used as the input to the temporal proposal subnet.

The potential action segments are initially by anchor segments. The anchor segments are the segments that are uniformly distributed throughout the input with different pre-defined scales. A 3D convolutional filter is used on the top of conv5d to extend the temporal receptive field for the anchor segments. Then, a 3D max-pooling filter is used to down-sample the spatial feature to produce a temporal-only feature map. The output of the 3D max-pooling layer is used as the feature vector to predict a relative offset to the central location and the length of each anchor segment by adding two convolutional layers.

There are three main functions: (1) selecting proposal segments using a greedy non-maximum suppression strategy to eliminate highly overlapping and low-confidence proposals; (2) extracting fixed-size features from selected proposals using 3D region-of-interest pooling; (3) activity classification and boundary regression for the selected proposal using the pooled features from a series of two fully connected layers.

Both the classification and regression subnets are optimized by the objective function:(2)$$Loss=\frac{1}{N_{cls}}\underset{i}{\sum }L_{cls}\left(a_{i},a_{i}^{*}\right)+\lambda \frac{1}{N_{reg}}\underset{i}{\sum }a_{i}^{*}L_{reg}\left(t_{i},t_{i}^{*}\right)\;$$

The SoftMax loss function is used for classification and the smooth L1 loss function is used for regression. The notation is explained in Table 1. The window regression and coordinate transformation are calculated using the equations below:(3)$$t_{x}=\frac{x-x_{a}}{w_{a}},\;t_{w}=\log\left(\frac{w}{w_{a}}\right)$$
(4)$$t_{x}^{*}=\frac{x^{*}-x_{a}}{w_{a}},\;t_{w}^{*}=log\left(w^{*}/w_{a}\right)$$
where *x* is the predicted window, *x_a_* is the anchor window, and *x** is the ground truth window.

##### Region Hierarchical Co-Occurrence Network

Temporal features are important for recognizing the underlying actions. A temporal representation of a skeleton’s motion was computed and explicitly fed into the network. For the skeleton of a person in frame t, this is formulated as $S^{t}=\left\{J_{1}^{t},\;J_{2}^{t},\;\ldots ,\;J_{N}^{t}\right\}$, where N is the number of joints and J=(x, y, z) is a 3D joint coordinate. The skeleton’s motion is defined as the temporal difference of each joint between two consecutive frames:(5)$$M^{t}=S^{t+1}-S^{t}=\left\{J_{1}^{t+1}-J_{1}^{t},\;\;J_{2}^{t+1}-J_{2}^{t},\;\ldots ,\;J_{N}^{t+1}-J_{N}^{t}\right\}$$

The hierarchical co-occurrence network [65,67] was employed to learn the joint co-occurrences and the temporal features jointly. The inputs are a skeleton sequence X with dimensions T × N × D and its corresponding skeletal motion with the same shape as X. They are fed into two point-level learning layers, since the kernel sizes along the joint dimensions are forced to 1 to learn the point-level representation of each joint. The transform layer switches the joint dimension with the coordinate dimension. The global co-occurrence features from all joints are extracted, and the features from the joint and motion are concatenated. Finally, the feature maps are flattened and further features are extracted by two fully connected layers.

The region hierarchical co-occurrence network, as shown in Figure 4, extracts the spatial–temporal features from the input joint sequence. The temporal proposal subnet and the action classification subnet used in the R-C3D network were employed to perform action recognition and detection processes.

#### 2.2.5. Ensemble Network Action Recognition

The goal of the action recognition and detection system is to recognize the clinically relevant actions and segment the detected actions out. Specifically, the per-frame action label needs to be generated. The ensemble network consists of three networks. All of them output the start frame, end frame, and predicted label of a detected segment. The per-frame label can be easily generated using the outputs of the networks. As a result, for each frame, three labels were given by the three networks in the ensemble network.

The final per-frame label was fused using the labels predicted by the three separate networks. If two of the networks voted the same action for a frame, the corresponding label of that frame was assigned to an action label with the most vote counts. Otherwise, the frame was considered as “none of the above”.

## 3. Results

### 3.1. Participants

Sixteen participants enrolled in the study and had the Foresite depth sensor installed in their home for a period of 3 months. The participants were 63.38 ± 12.84 years old. Nine participants identified as female. For each participant, a large volume of data was generated and collected by the depth sensor (average size = 3 Gb per day). The data labeling began as the data were collected, starting with participant 1. As of the time of writing, data from participants 1–5 had been labeled. Participant 10 had an ideal setup for the depth sensor within their home. The amount of data from these six participants totaled over 1 terabyte. Therefore, for the purposes of these analyses, only labeled data from participants 1–5 and 10 were used.

### 3.2. Algorithm Training and Testing

#### 3.2.1. CDC

To feed the video sequences to the network while maximizing the memory usage, a 16-frame sliding window was applied to segment the videos without overlapping. The CDC network was initialized by the model trained on the simulated kitchen dataset. The stochastic gradient decent was applied for optimization. Following conventional settings, we set the momentum to 0.9 and the weight decay to 0.005.

Eighty percent of the data were selected as the training set. We checked for balance using ground truth labeling to ensure that there was balance across action classes in the training and test sets. The best test results were generated with a learning rate of 0.001 and decreased by 0.1 for each 5000 iterations. The per-frame and per-action precision rates were 0.859 and 0.871, respectively.

#### 3.2.2. R-C3D

The same training strategy was used to train the R-C3D models for both the stroke dataset and the simulated kitchen dataset. For training on the stroke dataset, ten anchor segments were chosen. The scale values were: 2, 4, 5, 6, 8, 9, 10, 12, 14, and 16. To take advantage of the previous machine learning based on the healthy subjects [51], the model’s trainable parameters were initialized by the model trained on the simulated kitchen dataset.

For the training and test stroke dataset, 80% of the data were randomly selected for training and the rest of the data were used for testing. The different combinations of hyperparameter were investigated. The optimal results were generated, whereby the learning rate was 1 × 10^−14^, the learning rate weight decay was 0.0005, the learning rate decay step was 7, and the maximum number of epochs was set to 10. The per-action and per-frame precision rates were 0.86 and 0.88, respectively.

#### 3.2.3. R-HCN

The anchor segments initially had scales of [50,100,200,400] in the temporal proposal. The stroke dataset was split for training and testing. Eighty percent of the data were selected as the training set. The best test result was generated with the learning rate 1 × 10^−14^, and the learning weight decay was 0.0005. Since skeletal joint data are more sensitive to noise, only detected walking actions were considered. The per-action and per-frame precision rates were 0.74 and 0.72, respectively.

#### 3.2.4. Ensemble Network

The performance of the model with the highest accuracy rates in Table 2 was evaluated using the test set. The confusion matrix of the ensemble network on the test set is shown in Figure 5. The per-frame precision and per-action precision rates were 0.819 and 0.838 on the test set, respectively.

## 4. Discussion

The purpose of this study was to develop, train, and test an algorithm for activity recognition in people post-stroke in the home setting. The algorithm was found to be accurate at discerning six different actions. Future studies are planned to integrate the developed algorithm into a comprehensive sensor system deployed in the home setting of individuals post-stroke, as well as for older adults. These points are discussed below along with the study’s limitations.

The algorithm was able to successfully identify and discern six different actions. Although a rich set of approaches for action recognition and temporal action localization have been proposed, most of them require RGB datasets, meaning the privacy of patients cannot be preserved [37,38,39,40,41,42,43,44,45,46,47,48,49,50,51,52,53,54,55,56,57,61,66,67]. The existing algorithms utilizing depth data can only detect pre-designed actions of healthy individuals performed in well-controlled laboratory environments. Our proposed system temporally localized clinically relevant actions of stroke individuals using realistic in-home depth data. This followed existing work in the rehabilitation and specifically stroke rehabilitation literature, which has primarily focused on wearable sensors for activity recognition [68,69,70]. Several recent reviews and surveys of the field have suggested the use of computer vision and machine learning for activity recognition in rehabilitation [71,72]. However, most experiments and research have used healthy populations and existing datasets in controlled environments [73]. This is the first study to use real-world data from a clinical population to develop activity recognition algorithms. Our accuracy was good given the nature of the data. All upper extremity activities have limitations in terms of recognition due to the nature of the freedom of movement at the shoulder (6 degrees). This is a comparable challenge when using wearables for activity or exercise recognition [74].

The eventual goal Is to integrate the action recognition algorithm into the larger sensor system that was used in our prior work. The full sensor system consists of the same Foresite Healthcare depth sensor, motion sensors, and a bed sensor (under the mattress). The sensor system has been deployed in several clinical and community settings, including assisted living facilities and nursing homes, independent living apartments, community-based senior apartments, and rural community homes. The sensor system algorithms provide health alerts and messages to clinical staff, the resident, and their trusted support. The activity recognition algorithm developed in this study can be integrated into larger sensor system algorithms and provide additional detail on individuals’ overall health and performance. We also plan to further refine the algorithm using data from older adults with and without disabilities. Lastly, several of our ongoing studies integrate data from wearable sensors. Combined with the activity recognition algorithm, this could provide a more robust assessment of upper extremity activity and overall participation both in and out of the home.

This study has a few limitations. First, as with any technology-based research study, there is an inherent selection bias. The study participants had to agree to have the depth sensor placed in their home for three months. During recruitment, we contacted several people who met the study inclusion criteria but did not want to participate. The most common reason for not wanting to participate was that the individual did not want to have the depth sensor installed in their home. Some stated that they felt uncomfortable, and others did not provide an additional rationale. Trust is of the utmost importance in clinical practice and research, and it is crucial that individuals always feel comfortable and in control of their treatment and data. The second limitation in this study is that we have not yet compared the algorithm accuracy to other existing approaches. We plan to do so and report the findings in a future manuscript.

## 5. Conclusions

In conclusion, we have demonstrated that it is possible to develop, train, and test an algorithm for upper extremity activity recognition in individuals post-stroke. The algorithm can discern six different activities with good to excellent per-frame and per-action accuracy rates. Further, we were able to generate a very large amount of data from only 17 study participants (data from six participants were used for our analyses). This is one of the first studies to collect data for algorithm development, training, and testing in natural in-home environments with a clinical population. Our future work will further develop and assess the algorithm and pair the algorithm with other technologies, such as wearable sensors. This work has the potential to provide a novel, externally valid assessment of in-home activity post-stroke.

## Figures and Tables

**Figure 1 sensors-23-07872-f001:**
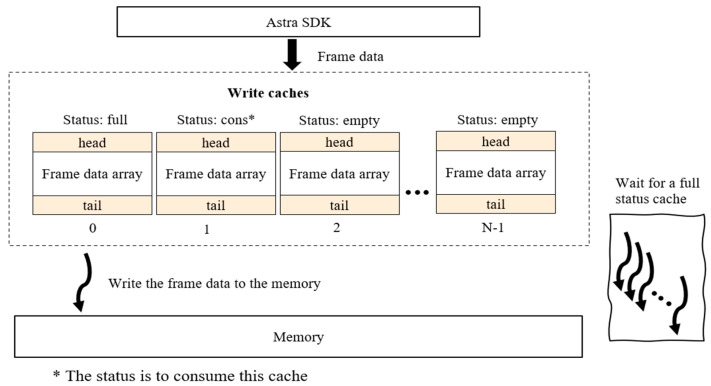
Data collection optimization mechanism. Multiple caches were allocated to store the income data frames. Several threads were created to store the data in batches whenever a cache is full. The frame rate with the optimization mechanism has been increased to 8 fps.

**Figure 2 sensors-23-07872-f002:**
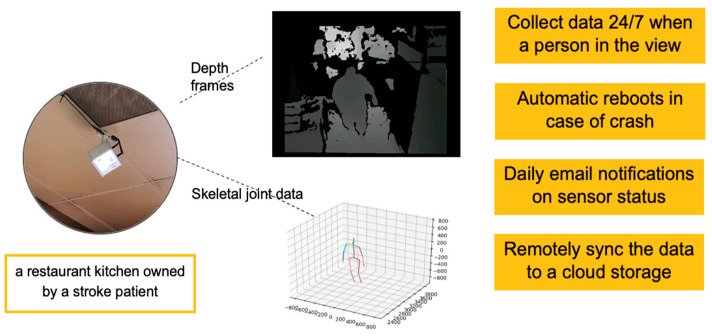
The main functionalities of the Foresite action logger system.

**Figure 3 sensors-23-07872-f003:**
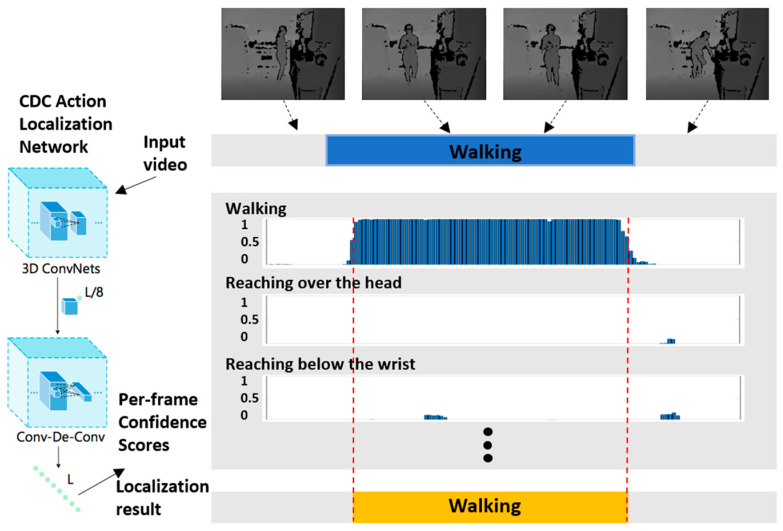
A framework for the positioning of temporal action recognition and localization. The color variations in the histogram represent per-frame confidence scores.

**Figure 4 sensors-23-07872-f004:**
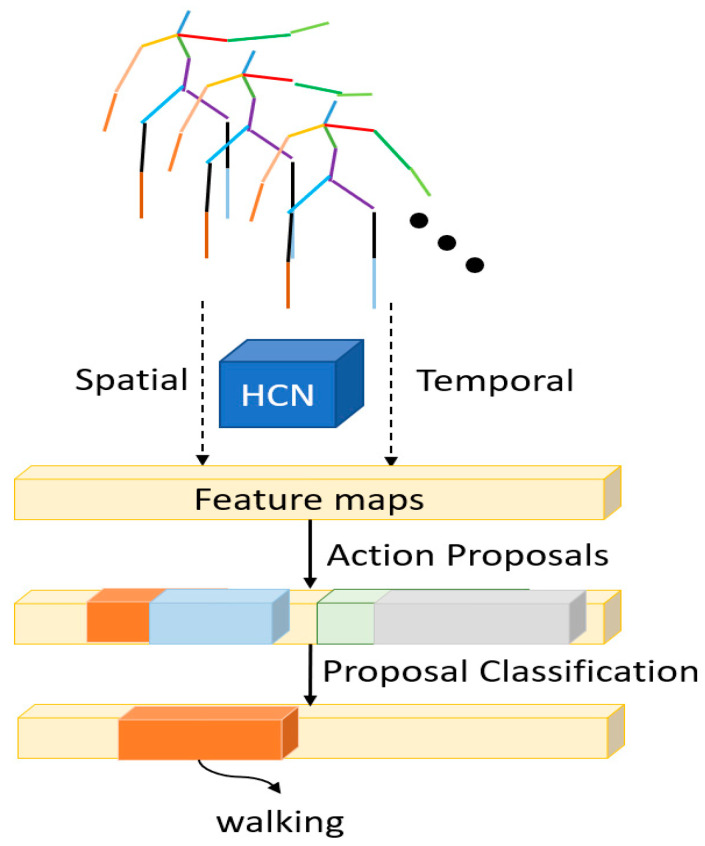
The architecture of the proposed region hierarchical co-occurrence network. The skeletal sequences at the top (multicolored figures), including spatial and temporal features are fed through the feature maps. The action proposals (orange and light blue) from the network are then flattened or cropped with the final proposal classification (longer orange bar on the bottom).

**Figure 5 sensors-23-07872-f005:**
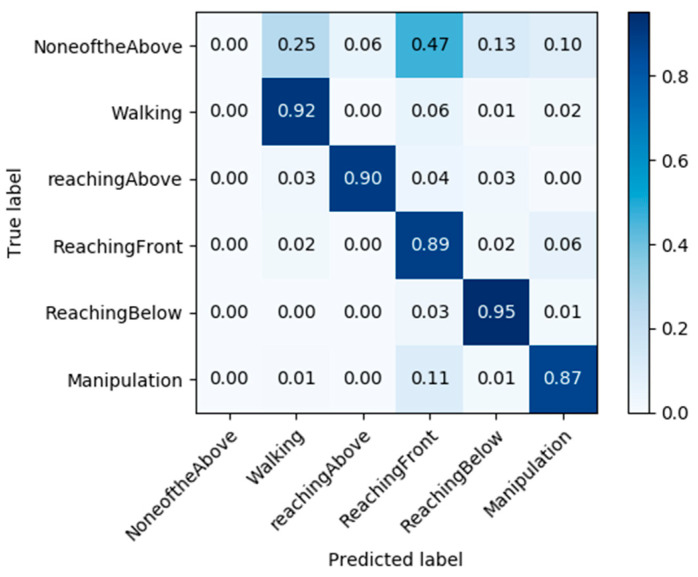
Normalized confusion matrix for recognizing actions from the ensemble network on the validation stroke dataset.

**Table 1 sensors-23-07872-t001:** The notations for the optimization function, where *i* is the anchor or proposal segment index in a batch and lambda is the trade-off parameter.

Classification	Regression
*N_cls_*: number of batches	*N_reg_:* number of anchor or proposal segments
*a_i_*: the predicted probability of the proposal or activity	*t_i_* = {*t_x_*,*t_w_*}: predicted relative offset to anchor segments or proposals
*a_i_**: the ground truth (1 if the anchor is positive and 0 if the anchor is negative.)	*t_i_**={*t_x_**,*t_w_**}: the coordinate transformation of ground truth segments to anchor segments or proposals

**Table 2 sensors-23-07872-t002:** Per-frame and per-action precision rates of the test videos from the ensemble network on the stroke dataset.

Dataset	Precision	Trial1	Trial2	Trial3	Mean	Std.
P1–3 *	Per Frame	0.785	0.791	0.679	0.752	0.063
	Per Action	0.831	0.801	0.761	0.798	0.035
P1–4	Per Frame	0.790	0.772	0.789	0.784	0.010
	Per Action	0.801	0.792	0.724	0.773	0.042
P1–5	Per Frame	0.818	0.817	0.742	0.792	0.044
	Per Action	0.839	0.801	0.791	0.810	0.025
P1–5, 10 ^	Per Frame	0.824	0.904	0.881	0.869	0.041
	Per Action	0.827	0.911	0.867	0.868	0.042

* P1–3: The test dataset contained the data for participants 1, 2, and 3. ^^^ P1–5,10: The test dataset contained the data from participants 1, 2, 3, 4, 5, and 10.

## Data Availability

Data available upon request from the authors. Given the sensitive nature of the depth sensor, the data are only available upon request with appropriate human subject assurances in place.

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
