# Peer review of "Development and Testing of a Daily Activity Recognition System for Post-Stroke Rehabilitation"

_sensors, 2023, doi:10.3390/s23187872_

Round 1

Reviewer 1 Report

Authors present analysis of data from a depth sensor in the home of patients with stroke.  They build 3 classifiers that predict several action categories with good precision.  The results are of importance given the growing emphasis on monitoring the effects of rehabilitation interventions in the home and community.  They overcome the privacy concerns related to video in the home by using a depth-only sensor mounted in the home.   The results are very promising, however details of the work need to be clarified.

11) In Section 2.2.2.  Please specify how much of the data (minutes, frames, etc.) was labeled by the human and subsequently used in the modeling.  Also, more explanation is needed on the annotation procedure.  How did annotators define the start and stop of every action category?   For example, is the reaching period defined as when the arm is moving forward and stop as soon as the arm begins to return to rest?  How are activities that combine categories scored?  For instance, reaching forward while holding an object?  What about walking while reaching?  Walking while holding an object?

22)  The paper states that only depth images are collected.  Please provide samples of these images.  While human annotation should be possible without an RGB image, it would be helpful to provide examples of the raw data. 

33)      Table 2 is confusing.  Please specify what the dataset names refer to.  The text implies data from 6 subjects are available, but there are only 4 datasets used here.    Why are 2 datasets combined in the last row?  Also, only precision is reported.  Authors should also report a measure of recall.  The meaning of the 3 trials needs to be clarified.  How is the “per Action” prediction compared to the true per frame labels from annotators?  Since the algorithm determines the start and stop points of each action, does an accurate prediction require all of the frames within this period to have the correct label?

44)      Can the algorithm identify different people in the images?  This would seem like an important feature to enable data collection from specific target individuals, when others can also use the space and be in view.

55)      The confusion matrix in Figure 6 implies that NoneoftheAbove class is never predicted by the algorithm. Please explain.

66)      It is unclear if all of the data is pooled and a single model is developed, or if several subject-specific models are developed.   Table 2 suggests subject-specific models, while Fig. 6 suggests a single model across all subjects was developed.   If the results are for subject-specific modeling, how do researchers plan to deal with the need for a human annotator to label data.

77)      They used a random 80/20 split in the data for training and testing.  Was any there any attempt to balance the data in the test set so that the ratio of classes in the test data was representative of the entire dataset?

Reviewer 2 Report

Pros:

Pointed out the importance of untrimmed temporal action labelling(segmentation)

Fusing three SOTA  for the best per-frame action labelling.

The authors collected their own 6-subjects dataset, a very interesting pilot study.

Cons:

For post-stroke rehabilitation, using sensing and ML technologies to monitor patients’ behaviour can provide an objective measurement. There are different sensing techniques, and video-based approaches, suffer from several issues such as privacy, non-continuous-measurement (for fixed cameras), in contrast to wearable-based approaches. A good justification should be provided why video-based, instead of other sensing modalities.

Daily activity data were collected from seventeen post-stroke participants’ homes (line 17) over three months. It is nice to see such longitudinal data, yet it is sixteen (line 419) participants, and then six (line 424) participants.  Putting 17 participants in abstract is very misleading---- since only 6 participants data were used.

In abstract, why validation set’s results were reported, which was normally used for tune hyper-parameters.  There should be performance of test set. What does these per-action precision mean to rehabilitation, it seems there is a gap between prediction system and application. In line 142, “In this paper, we focus on the daily activity recognition methods and results, the assessment component is described elsewhere [50]”------ however, when the title/abstract is about the post-stroke rehab scenario---- it would be better to discuss the link between activity recognition and rehab, and give some rehab assessment results.

In home environment, there might be conditions when patients are far away from camera or in different rooms. Does such information missing affect rehab assessment? Action labelling only couldn’t answer this question since it assumes action data is acquired already.  

Missing other baselines. Is the data balanced, how to address the data imbalanced problem (e.g., is class “nonoftheabove” the majority class? ). If it is real-world dataset as claimed in line 481 ---- I guess the DL architecture should be totally different since the other five activity classes are very rare.

Line 434 “the training dataset was formed yb 80% of the training and test set and the remaining data was used as the test set”------ not clear which one is the test set?

Line 435 “the best test results were generated with the learning rate as 0.001…” seems like the hyper-parameters are tuned based on the test set, which is not allowed.   Hyper-parameters should be tuned using validation set.

It would be good to provide a table detailing the experimental protocols.

Reviewer 3 Report

In line 146 the authors says "There are two main parts of the daily activity recognition system". But, they list three: "the action data logging system", "the action recognition"  and the "temporal action localization components".

As was done in Section 2.2.3, I believe it would be important to use a figure or flowchart to explain the algorithms and systems in Sections 2.1, 2.2, and 2.2.2.

The work made a very detailed description of its methods. However its results are presented in a very succinct way. I believe that section 2.2.4 and the following sections are already part of the results of the work. These sections demonstrate how the employed methods were used to arrive at the proposed system.

The discussion of the results is intertwined with the conclusion, even citing what are the future works of the research.

Reviewer 4 Report

Good submission, minor revisions can be seen below;

-Explicit contributions to literature as well as operating hypothesis needs to be fleshed out in the introduction

-Any word on ethical approval?

-Worth clarifying if the study was done with all healthy participants?

-The methods are there but need to be fleshed out satisfactorily enough

-Satisfactory discussions, but a conclusion and future work need to be fleshed out in subsequent revision

n/a

Round 2

Reviewer 1 Report

Authors have addressed my concerns.  

Reviewer 2 Report

The authors have addressed most of the concerns I have, and I think the work can be accepted. Although the test scenarios are very ideal, this work can be deemed as a pilot study, and give readers some takeaways.